# Manifold-based Similarity Adaptation for Label Propagation

**Masayuki Karasuyama and Hiroshi Mamitsuka**
Bioionformatics Center, Institute for Chemical Research, Kyoto University, Japan
{karasuyama,mami}@kuicr.kyoto-u.ac.jp

## Abstract

Label propagation is one of the state-of-the-art methods for semi-supervised learning, which estimates labels by propagating label information through a graph. Label propagation assumes that data points (nodes) connected in a graph should have similar labels. Consequently, the label estimation heavily depends on edge weights in a graph which represent similarity of each node pair. We propose a method for a graph to capture the manifold structure of input features using edge weights parameterized by a similarity function. In this approach, edge weights represent both similarity and local reconstruction weight simultaneously, both being reasonable for label propagation. For further justification, we provide analytical considerations including an interpretation as a cross-validation of a propagation model in the feature space, and an error analysis based on a low dimensional manifold model. Experimental results demonstrated the effectiveness of our approach both in synthetic and real datasets.

## 1 Introduction

Graph-based learning algorithms have received considerable attention in machine learning community. For example, *label propagation* (e.g., [1, 2]) is widely accepted as a state-of-the-art approach for semi-supervised learning, in which node labels are estimated through the input graph structure. A common important property of these graph-based approaches is that the *manifold* structure of the input data can be captured by the graph. Their practical performance advantage has been demonstrated in various application areas.

On the other hand, it is well-known that the accuracy of the graph-based methods highly depends on the quality of the input graph (e.g., [1, 3–5]), which is typically generated from a set of numerical input vectors (i.e., feature vectors). A general framework of graph-based learning can be represented as the following *three-step procedure*:

**Step 1:** Generating graph edges from given data, where nodes of the generated graph correspond to the instances of input data.
**Step 2:** Giving weights to the graph edges.
**Step 3:** Estimating node labels based on the generated graph, which is often represented as an *adjacency matrix*.

In this paper, we focus on the second step in the three-step procedure; estimating edge weights for the subsequent label estimation. Optimizing edge weights is difficult in semi-supervised learning, because there are only a small number of labeled instances. Also this problem is important because edge weights heavily affect final prediction accuracy of graph-based methods, while in reality rather simple heuristics strategies have been employed.

There are two standard approaches for estimating edge weights: similarity function based- and *locally linear embedding* (LLE) [6] based-approaches. Each of these two approaches has its own

disadvantage. The similarity based approaches use similarity functions, such as Gaussian kernel, while most similarity functions have tuning parameters (such as the width parameter of Gaussian kernel) that are in general difficult to be tuned. On the other hand, in LLE, the true underlying manifold can be approximated by a graph which minimizes a local reconstruction error. LLE is more sophisticated than the similarity-based approach, and LLE based graphs have been applied to semi-supervised learning [5, 7–9]. However LLE is noise-sensitive [10]. In addition, to avoid a kind of degeneracy problem [11], LLE has to have additional tuning parameters.

Our approach is a similarity-based method, yet also captures the manifold structure of the input data; we refer to our approach as *adaptive edge weighting* (AEW). In AEW, graph edges are determined by a data adaptive manner in terms of both similarity and manifold structure. The objective function in AEW is based on local reconstruction, by which estimated weights capture the manifold structure, where each edge is parameterized as a similarity function of each node pair. Consequently, in spite of its simplicity, AEW has the following three advantages:

- Compared to LLE based approaches, our formulation alleviates the problem of over-fitting due to the parameterization of weights. In our experiments, we observed that AEW is robust against noise of input data using synthetic data set, and we also show the performance advantage of AEW in eight real-world datasets.

- Similarity based representation of edge weights is reasonable for label propagation because transitions of labels are determined by those weights, and edge weights obtained by LLE approaches may not represent node similarity.

- AEW does not have additional tuning parameters such as regularization parameters. Although the number of edges in a graph cannot be determined by AEW, we show that performance of AEW is robust against the number of edges compared to standard heuristics and a LLE based approach.

We provide further justifications for our approach based on the ideas of *feature propagation* and *local linear approximation*. Our objective function can be seen as a cross validation error of a propagation model for feature vectors, which we call feature propagation. This allows us to interpret that AEW optimizes graph weights through cross validation (for prediction) in the feature vector space instead of label space, assuming that input feature vectors and given labels share the same local structure. Another interpretation is provided through local linear approximation, by which we can analyze the error of local reconstruction in the output (label) space under the assumption of low dimensional manifold model.

## 2 Graph-based Semi-supervised Learning

In this paper we use label propagation, which is one of the state-of-the-art graph-based learning algorithms, as the methods in the third step in the three-step procedure. Suppose that we have $n$ feature vectors $\mathcal{X} = \{\boldsymbol{x}_1, \ldots, \boldsymbol{x}_n\}$, where $\boldsymbol{x}_i \in \mathbb{R}^p$. An undirected graph $\mathcal{G}$ is generated from $\mathcal{X}$, where each node (or vertex) corresponds to each data point $\boldsymbol{x}_i$. The graph $\mathcal{G}$ can be represented by the adjacency matrix $\boldsymbol{W} \in \mathbb{R}^{n \times n}$ where $(i, j)$-element $W_{ij}$ is a weight of the edge between $\boldsymbol{x}_i$ and $\boldsymbol{x}_j$. The key idea of graph-based algorithms is so-called *manifold assumption*, in which instances connected by large weights $W_{ij}$ on a graph have similar labels (meaning that labels smoothly change on the graph).

For the adjacency matrix $W_{ij}$, the following weighted *k-nearest neighbor* (k-NN) graph is commonly used in graph-based learning algorithms [1]:

$$W_{ij} = \begin{cases} \exp\left(-\sum_{d=1}^{p} \frac{(x_{id}-x_{jd})^2}{\sigma_d^2}\right), & j \in \mathcal{N}_i \text{ or } i \in \mathcal{N}_j, \\ 0, & \text{otherwise,} \end{cases} \tag{1}$$

where $x_{id}$ is the $d$-th element of $\boldsymbol{x}_i$, $\mathcal{N}_i$ is a set of indices of the $k$-NN of $\boldsymbol{x}_i$, and $\{\sigma_d\}_{d=1}^{p}$ is a set of parameters. [1] shows this weighting can also be interpreted as the solution of the heat equation on the graph.

From this adjacency matrix, the *graph Laplacian* can be defined by

$$\boldsymbol{L} = \boldsymbol{D} - \boldsymbol{W},$$

where $D$ is a diagonal matrix with the diagonal entry $D_{ii} = \sum_j W_{ij}$. Instead of $L$, normalized variants of Laplacian such as $L = I - D^{-1}W$ or $L = I - D^{-1/2}WD^{-1/2}$ is also used, where $I \in \mathbb{R}^{n \times n}$ is the identity matrix.

Among several label propagation algorithms, we mainly use the formulation by [1], which is the standard formulation of graph-based semi-supervised learning. Suppose that the first $\ell$ data points in $\mathcal{X}$ are labeled by $\mathcal{Y} = \{y_1, \ldots, y_\ell\}$, where $y_i \in \{1, \ldots, c\}$ and $c$ is the number of classes. The goal of label propagation is to predict the labels of unlabeled nodes $\{x_{\ell+1}, \ldots, x_n\}$. The scoring matrix $F$ gives an estimation of the label of $x_i$ by $\operatorname{argmax}_j F_{ij}$. Label propagation can be defined as estimating $F$ in such a way that score $F$ smoothly changes on a given graph as well as it can predict given labeled points. The following is standard formulation, which is called the *harmonic Gaussian field* (HGF) model, of label propagation [1]:

$$\min_F \operatorname{trace}\left(F^\top L F\right) \quad \text{subject to} \ F_{ij} = Y_{ij}, \text{ for } i = 1, \ldots, \ell.$$

where $Y_{ij}$ is the label matrix with $Y_{ij} = 1$ if $x_i$ is labeled as $y_i = j$; otherwise, $Y_{ij} = 0$, In this formulation, the scores for labeled nodes are fixed as constants. This formulation can be reduced to linear systems, which can be solved efficiently, especially when Laplacian $L$ has some sparse structure.

## 3 Basic Framework of Proposed Approach

The performance of label propagation heavily depends on quality of an input graph. Our proposed approach, *adaptive edge weighting* (AEW), optimizes edge weights for the graph-based learning algorithms. We note that AEW is for the second step of the three step procedure and has nothing to do with the first and third steps, meaning that any methods in the first and third steps can be combined with AEW. In this paper we consider that the input graph is generated by $k$-NN graph (the first step is based on $k$-NN), while we note that AEW can be applied to any types of graphs.

First of all, graph edges should satisfy the following conditions:

- Capturing the manifold structure of the input space.
- Representing similarity between two nodes.

These two conditions are closely related to *manifold assumption* of graph-based learning algorithms, in which labels vary smoothly along the input manifold. Since the manifold structure of the input data is unknown beforehand, the graph is used to approximate the manifold (the first condition). Subsequent predictions are performed in such a way that labels smoothly change according to the similarity structure provided by the graph (the second condition). Our algorithm simultaneously pursues these two important aspects of the graph for the graph-based learning algorithms.

We define $W_{ij}$ as a similarity function of two nodes (1), using Gaussian kernel in this paper (Note: other similarity functions can also be used). We estimate $\sigma_d$ so that the graph represents manifold structure of the input data by the following optimization problem:

$$\min_{\{\sigma_d\}_{d=1}^p} \sum_{i=1}^n \|x_i - \frac{1}{D_{ii}} \sum_{j \sim i} W_{ij} x_j\|_2^2, \tag{2}$$

where $j \sim i$ means that $j$ is connected to $i$. This minimizes the reconstruction error by local linear approximation, which captures the input manifold structure, in terms of the parameters of the similarity function. We will describe the motivation and analytical properties of the objective function in Section 4. We further describe advantages of this function over existing approaches including well-known *locally linear embedding* (LLE) [6] based methods in Section 5, respectively.

To optimize (2), we can use any gradient-based algorithm such as steepest descent and conjugate gradient (in the later experiments, we used steepest descent method). Due to the non-convexity of the objective function, we cannot guarantee that solutions converge to the global optimal which means that the solutions depend on the initial $\sigma_d$. In our experiments, we employed well-known median heuristics (e.g., [12]) for setting initial values of $\sigma_d$ (Section 6). Another possible strategy is to use a number of different initial values for $\sigma_d$, which needs a high computational cost. The

gradient can be computed efficiently, due to the sparsity of the adjacency matrix. Since the number of edges of a $k$-NN graph is $O(nk)$, the derivative of adjacency matrix $\boldsymbol{W}$ can be calculated by $O(nkp)$. Then the entire derivative of the objective function can be calculated by $O(nkp^2)$. Note that $k$ often takes a small value such as $k = 10$.

## 4  Analytical Considerations

In Section 3, we defined our approach as the minimization of the local reconstruction error of input features. We describe several interesting properties and interpretations of this definition.

### 4.1  Derivation from Feature Propagation

First, we show that our objective function can be interpreted as a cross-validation error of the HGF model for the feature vector $\boldsymbol{x}$ on the graph. Let us divide a set of node indices $\{1, \ldots, n\}$ into a training set $\mathcal{T}$ and a validation set $\mathcal{V}$. Suppose that we try to predict $\boldsymbol{x}$ in the validation set $\{\boldsymbol{x}_i\}_{i \in \mathcal{V}}$ from the given training set $\{\boldsymbol{x}_i\}_{i \in \mathcal{T}}$ and the adjacency matrix $\boldsymbol{W}$. For this prediction problem, we consider the HGF model for $\boldsymbol{x}$:

$$\min_{\hat{\boldsymbol{X}}} \operatorname{trace}\left(\hat{\boldsymbol{X}}^{\top} \boldsymbol{L} \hat{\boldsymbol{X}}\right) \quad \text{subject to } \hat{x}_{ij} = x_{ij}, \text{ for } i \in \mathcal{T},$$

where $\boldsymbol{X} = (\boldsymbol{x}_1, \boldsymbol{x}_2, \ldots \boldsymbol{x}_n)^{\top}$, $\hat{\boldsymbol{X}} = (\hat{\boldsymbol{x}}_1, \hat{\boldsymbol{x}}_2, \ldots \hat{\boldsymbol{x}}_n)^{\top}$, and $x_{ij}$ and $\hat{x}_{ij}$ indicate $(i, j)$-th entries of $\boldsymbol{X}$ and $\hat{\boldsymbol{X}}$ respectively. In this formulation, $\hat{\boldsymbol{x}}_i$ corresponds to a prediction for $\boldsymbol{x}_i$. Note that only $\hat{\boldsymbol{x}}_i$ in the validation set $\mathcal{V}$ is regarded as free variables in the optimization problem because the other $\{\hat{\boldsymbol{x}}_i\}_{i \in \mathcal{T}}$ is fixed at the observed values by the constraint. This can be interpreted as propagating $\{\boldsymbol{x}_i\}_{i \in \mathcal{T}}$ to predict $\{\boldsymbol{x}_i\}_{i \in \mathcal{V}}$. We call this process as *feature propagation*.

When we employ leave-one-out as the cross-validation of the feature propagation model, we obtain

$$\sum_{i=1}^{n} \|\boldsymbol{x}_i - \hat{\boldsymbol{x}}_{-i}\|_2^2, \tag{3}$$

where $\hat{\boldsymbol{x}}_{-i}$ is a prediction for $\boldsymbol{x}_i$ with $\mathcal{T} = \{1, \ldots, i-1, i+1, \ldots, n\}$ and $\mathcal{V} = \{i\}$. Due to the local averaging property of HGF [1], we see $\hat{\boldsymbol{x}}_{-i} = \sum_j W_{ij} \boldsymbol{x}_j / D_{ii}$, and then (3) is equivalent to our objective function (2). From this equivalence, AEW can be interpreted as the parameter optimization in graph weights of the HGF model for feature vectors through the leave-one-out cross-validation. This also means that our framework estimates labels using the adjacency matrix $\boldsymbol{W}$ optimized in the feature space instead of the output (label) space. Thus, if input features and labels share the same adjacency matrix (i.e., sharing the same local structure), the minimization of the objective function (2) should estimate the adjacency matrix which accurately propagates the labels of graph nodes.

### 4.2  Local Linear Approximation

The feature propagation model provides the interpretation of our approach as the optimization of the adjacency matrix under the assumption that $x$ and $y$ can be reconstructed by the same adjacency matrix. We here justify this assumption in a more formal way from a viewpoint of local reconstruction with a lower dimensional manifold model.

As shown in [1], HGF can be regarded as local reconstruction methods, which means the prediction can be represented as weighted local averages:

$$F_{ik} = \frac{\sum_j W_{ij} F_{jk}}{D_{ii}} \quad \text{for } i = \ell + 1, \ldots, n.$$

We show the relationship between the local reconstruction error in the feature space described by our objective function (2) and the output space. For simplicity we consider the vector form of the score function $\boldsymbol{f} \in \mathbb{R}^n$ which can be considered as a special case of the score matrix $\boldsymbol{F}$, and discussions here can be applied to $\boldsymbol{F}$. The same analysis can be approximately applied to other graph learning methods such as local global consistency [2] because it has similar local averaging form as the above, though we omitted here.

We assume the following manifold model for the input feature space, in which $\boldsymbol{x}$ is generated from corresponding some lower dimensional variable $\boldsymbol{\tau} \in \mathbb{R}^q$: $\boldsymbol{x} = g(\boldsymbol{\tau}) + \varepsilon_x$, where $g : \mathbb{R}^q \to \mathbb{R}^p$ is a smooth function, and $\varepsilon_x \in \mathbb{R}^p$ represents noise. In this model, $y$ is also represented by some function form of $\boldsymbol{\tau}$: $y = h(\boldsymbol{\tau}) + \varepsilon_y$, where $h : \mathbb{R}^q \to \mathbb{R}$ is a smooth function, and $\varepsilon_y \in \mathbb{R}$ represents noise (for simplicity, we consider a continuous output rather than discrete labels). For this model, the following theorem shows the relationship between the reconstruction error of the feature vector $\boldsymbol{x}$ and the output $y$:

**Theorem 1.** *Suppose $\boldsymbol{x}_i$ can be approximated by its neighbors as follows*

$$\boldsymbol{x}_i = \frac{1}{D_{ii}} \sum_{j \sim i} W_{ij} \boldsymbol{x}_j + \boldsymbol{e}_i, \tag{4}$$

*where $\boldsymbol{e}_i \in \mathbb{R}^p$ represents an approximation error. Then, the same adjacency matrix reconstructs the output $y_i \in \mathbb{R}$ with the following error:*

$$y_i - \frac{1}{D_{ii}} \sum_{j \sim i} W_{ij} y_j = \boldsymbol{J} \boldsymbol{e}_i + O(\delta \boldsymbol{\tau}_i) + O(\varepsilon_x + \varepsilon_y), \tag{5}$$

*where $\boldsymbol{J} = \frac{\partial h(\boldsymbol{\tau}_i)}{\partial \boldsymbol{\tau}^\top} \left( \frac{\partial g(\boldsymbol{\tau}_i)}{\partial \boldsymbol{\tau}^\top} \right)^+$ with superscript $^+$ indicates pseudoinverse, and $\delta \boldsymbol{\tau}_i = \max_j(\|\boldsymbol{\tau}_i - \boldsymbol{\tau}_j\|_2^2)$.*

See our supplementary material for the proof of this theorem. From (5), we can see that the reconstruction error of $y_i$ consists of three terms. The first term includes the reconstruction error for $\boldsymbol{x}_i$ which is represented by $\boldsymbol{e}_i$, and the second term is the distance between $\boldsymbol{\tau}_i$ and $\{\boldsymbol{\tau}_j\}_{j \sim i}$. These two terms have a kind of trade-off relationship because we can reduce $\boldsymbol{e}_i$ if we use a lot of data points $\boldsymbol{x}_j$, but then $\delta \boldsymbol{\tau}_i$ would increase. The third term is the intrinsic noise which we cannot directly control. In spite of its importance, this simple relationship has not been focused on in the context of graph estimation for semi-supervised learning, in which a LLE based objective function has been used without clear justification [5, 7–9].

A simple approach to exploit this theorem would be a regularization formulation, which can be a minimization of a combination of the reconstruction error for $\boldsymbol{x}$ and a penalization term for distances between data points connected by edges. Regularized LLE [5, 8, 13, 14] can be interpreted as one realization of such an approach. However, in semi-supervised learning, selecting appropriate values of the regularization parameter is difficult. We therefore optimize edge weights through parameters of the similarity function, especially the bandwidth parameter of Gaussian similarity function $\sigma$. In this approach, a very large bandwidth (giving large weights to distant data points) may cause a large reconstruction error, while an extremely small bandwidth causes the problem of not giving enough weights to reconstruct.

For symmetric normalized graph Laplacian, we can not apply Theorem 1 to our algorithm. See supplementary material for modified version of Theorem 1 for normalized Laplacian. In the experiments, we also report results for normalized Laplacian and show that our approach can improve prediction accuracy as in the case of unnormalized Laplacian.

## 5   Related Topics

LLE [6] can also estimate graph edges based on a similar objective function, in which $\boldsymbol{W}$ is directly optimized as a real valued matrix. This manner has been used in many methods for graph-based semi-supervised learning and clustering [5, 7–9], but LLE is very noise-sensitive [10], and resulting weights $W_{ij}$ cannot necessarily represent similarity between the corresponding nodes $(i, j)$. For example, for two nearly identical points $\boldsymbol{x}_{j_1}$ and $\boldsymbol{x}_{j_2}$, both connecting to $\boldsymbol{x}_i$, it is not guaranteed that $W_{ij_1}$ and $W_{ij_2}$ have similar values. To solve this problem, a regularization term can be introduced [11], while it is not easy to optimize the regularization parameter for this term. On the other hand, we optimize parameters of the similarity (kernel) function. This parameterized form of edge weights can alleviate the over-fitting problem. Moreover, obviously, the optimized weights still represent the node similarity.

Although several model selection approaches (such as cross-validation and marginal likelihood maximization) have been applied to optimizing graph edge weights by regarding them as usual hyper-

parameters in supervised learning [3, 4, 15], most of them need labeled instances and become unreliable under the cases with few labels. Another approach is optimizing some criterion designed specifically for graph-based algorithms (e.g., [1, 16]). These criteria often have degenerate (trivial) solutions for which heuristics are used to prevent, but the validity of those heuristics is not clear. Compared to these approaches, our approach is more general and flexible for problem settings, because AEW is independent of the number of classes, the number of labels, and subsequent label estimation algorithms. In addition, model selection based approaches are basically for the third step in the three-step procedure, by which AEW can be combined with such methods, like that the optimized graph by AEW can be used as the input graph of these methods.

Besides $k$-NN, there have been several methods generating a graph (edges) from the feature vectors (e.g., [9, 17]). Our approach can also be applied to those graphs because AEW only optimizes weights of edges. In our experiments, we used the edges of the $k$-NN graph as the initial graph of AEW. We then observed that AEW is not sensitive to the choice of $k$, comparing with usual $k$-NN graphs. This is because the Gaussian similarity value becomes small if $x_i$ and $x_j$ are not close to each other to minimize the reconstruction error (2). In other words, redundant weights can be reduced drastically, because in the Gaussian kernel, weights decay exponentially according to the squared distance.

Metric learning is another approach to adapting similarity, while metric learning is not for graphs. A standard method for incorporating graph information into metric learning is to use some graph-based regularization, in which graph weights must be determined beforehand. For example, in [18], a graph is generated by LLE, of which we already described the disadvantages. Another approach is [19], which estimates a distance metric so that the k-NN graph in terms of the obtained metric should reproduce a given graph. This approach is however not for semi-supervised learning, and it is unclear if this approach works for semi-supervised settings. Overall metric learning is developed from a different context from our setting, by which it has not been justified that metric learning can be applied to label propagation.

## 6 Experiments

We evaluated the performance of our approach using synthetic and real-world datasets. We investigated the performance of AEW using the harmonic Gaussian field (HGF) model. For comparison, we used *linear neighborhood propagation* (LNP) [5], which generates a graph using a LLE based objective function. LNP can have two regularization parameters, one of which is for the LLE process (the first and second steps in the three-step procedure), and the other is for the label estimation process (the third step in the three-step procedure). For the parameter in the LLE process, we used the heuristics suggested by [11], and for the label propagation process, we chose the best parameter value in terms of the test accuracy. HGF does not have such hyper-parameters. All results were averaged over 30 runs with randomly sampled data points.

### 6.1 Synthetic datasets

We here use two datasets in Figure 1 having the same form, but Figure 1 (b) has several noisy data points which may become *bridge points* (points connecting different classes [5]). In both cases, the number of classes is 4 and each class has 100 data points (thus, $n = 400$).

Table 1 shows the error rates for the unlabeled nodes of HGF and LNP under 0-1 loss. For HGF, we used the median heuristics to choose the parameter $\sigma_d$ in similarity function (1), meaning that a common $\sigma$ ($= \sigma_1 = \ldots = \sigma_p$) is set as the median distance between all connected pairs of $x_i$. The symmetric normalized version of graph Laplacian was used. The optimization of AEW started from the median $\sigma_d$. The results by AEW are shown in the column 'AEW + HGF' of Table 1. The number of labeled nodes was 10 in each class ($\ell = 40$, i.e., 10% of the entire datasets), and the number of neighbors in the graphs was set as $k = 10$ or 20.

In Table 1, we see HGF with AEW achieved better prediction accuracies than the median heuristics and LNP in all cases. Moreover, for both of datasets (a) and (b), AEW was most robust against the change of the number of neighbors $k$. This is because $\sigma_d$ is automatically adjusted in such a way that the local reconstruction error is minimized and then weights for connections between

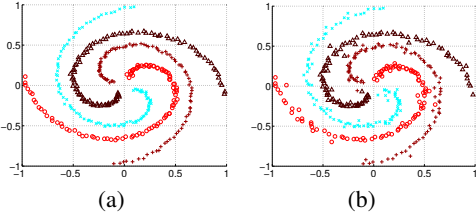

(a)　　　　　(b)

Figure 1: Synthetic datasets.

Table 1: Test error comparison for synthetic datasets. The best methods according to $t$-test with the significant level of $5\%$ are highlighted with boldface.

| data | $k$ | HGF | AEW + HGF | LNP |
|------|-----|------|-----------|------|
| (a) | 10 | .057 (.039) | **.020 (.027)** | .039 (.026) |
| (a) | 20 | .261 (.048) | **.020 (.028)** | .103 (.042) |
| (b) | 10 | .119 (.054) | **.073 (.035)** | .103 (.038) |
| (b) | 20 | .280 (.051) | **.077 (.035)** | .148 (.047) |

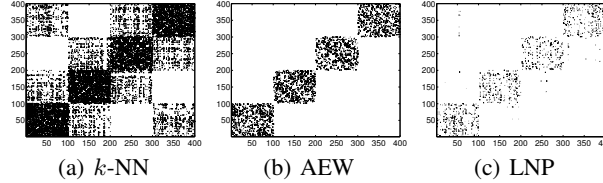

(a) $k$-NN　　　(b) AEW　　　(c) LNP

Figure 2: Resulting graphs for the synthetic dataset of Figure 1 (a) ($k = 20$).

different manifolds are reduced. Although LNP also minimizes the local reconstruction error, LNP may connect data points far from each other if it reduces the reconstruction error.

Figure 2 shows the graphs generated by (a) $k$-NN, (b) AEW, and (c) LNP, under $k = 20$ for the dataset of Figure 1 (a). In Figure 2, the $k$-NN graph connects a lot of nodes in different classes, while AEW favorably eliminates those undesirable edges. LNP also has less edges between different classes compared to $k$-NN, but it still connects different classes. AEW reveals the class structure more clearly, which can lead the better prediction performance of subsequent learning algorithms.

## 6.2 Real-world datasets

We examined the performance of our approach on the eight popular datasets shown in Table 2, namely COIL (COIL-20) [20], USPS (a preprocessed version from [21]), MNIST [22], ORL [23], Vowel [24], Yale (Yale Face Database B) [25], optdigit [24], and UMIST [26].

We evaluated two variants of the HGF model. In what follows, 'HGF' indicates HGF using unnormalized graph Laplacian $\boldsymbol{L} = \boldsymbol{D} - \boldsymbol{W}$, and 'N-HGF' indicates HGF using symmetric normalized Laplacian $\boldsymbol{L} = \boldsymbol{I} - \boldsymbol{D}^{-1/2}\boldsymbol{W}\boldsymbol{D}^{-1/2}$. For both of two variants, the me-

Table 2: List of datasets.

|  | $n$ | $p$ | # classes |
|--------|------|------|-----------|
| COIL | 500 | 256 | 10 |
| USPS | 1000 | 256 | 10 |
| MNIST | 1000 | 784 | 10 |
| ORL | 360 | 644 | 40 |
| Vowel | 792 | 10 | 11 |
| Yale | 250 | 1200 | 5 |
| optdigit | 1000 | 256 | 10 |
| UMIST | 518 | 644 | 20 |

dian heuristics was used to set $\sigma_d$. To adapt the difference of local scale, we here use local scaling kernel [27] as the similarity function. Figure 3 shows the test error for unlabeled nodes. In this figure, two dashed lines with different markers are by HGF and N-HGF, while two solid lines with the same markers are by HGF with AEW. The performance difference within the variants of HGF was not large, compared to the effect of AEW, particularly in COIL, ORL, Vowel, Yale, and UMIST. We can rather see that AEW substantially improved the prediction accuracy of HGF in most cases. LNP is by the solid line without any markers. LNP outperformed HGF (without AEW, shown as the dashed lines) in COIL, ORL, Vowel, Yale and UMIST, while HGF with AEW (at least one of three variants) achieved better performance than LNP in all these datasets except for Yale (In Yale, LNP and HGF with AEW attained a similar accuracy).

Overall AEW-N-HGF had the best prediction accuracy, where typical examples were USPS and MNIST. Although Theorem 1 exactly holds only for AEW-HGF, we can see that AEW-N-HGF, in which the degrees of the graph nodes are scaled by normalized Laplacian had highly stable performance.

We further examined the effect of $k$. Figure 4 shows the test error for $k = 20$ and 10, using N-HGF, AEW-N-HGF, and LNP for COIL dataset. The number of labeled instances is the midst value in

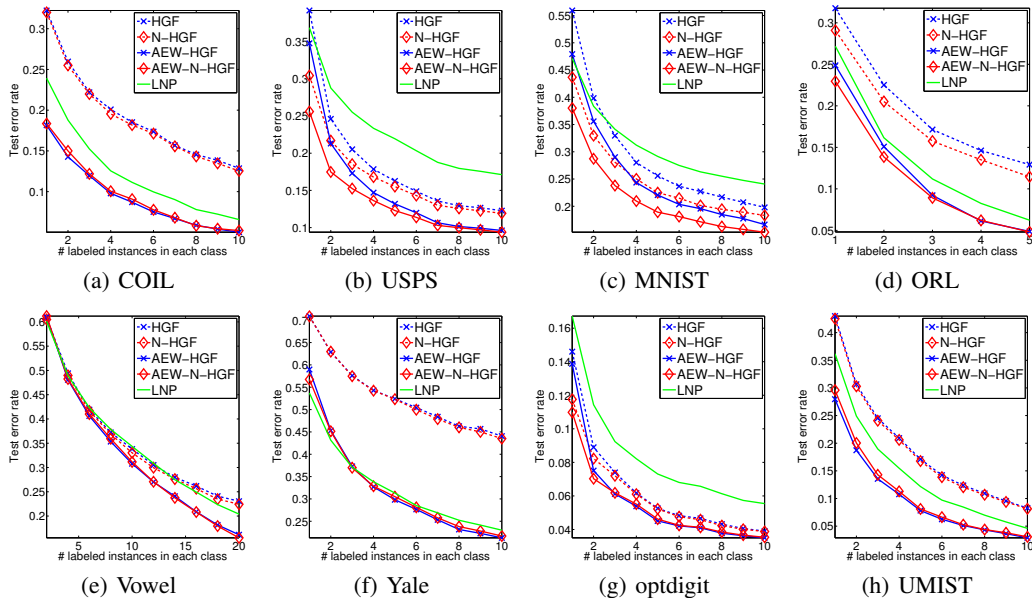

|            |            |            |            |
| :--------: | :--------: | :--------: | :--------: |
| (a) COIL   | (b) USPS   | (c) MNIST  | (d) ORL    |
| (e) Vowel  | (f) Yale   | (g) optdigit | (h) UMIST |

Figure 3: Performance comparison on real-world datasets. HGFs with AEW are by solid lines with markers, while HGFs with median heuristics is by dashed lines with the same markers, and LNP is by a solid line without any markers. For N-HGF and AWE-N-HGF, 'N' indicates normalized Laplacian.

the horizontal axis of Figure 3 (a) (5 in each class). We can see that the test error of AEW is not sensitive to $k$. Performance of N-HGF with $k = 20$ was worse than that with $k = 10$. On the other hand, AEW-N-HGF with $k = 20$ had a similar performance to that with $k = 10$.

## 7 Conclusions

We have proposed the *adaptive edge weighting* (AEW) method for graph-based semi-supervised learning. AEW is based on the local reconstruction with the constraint that each edge represents the similarity of each pair of nodes. Due to this constraint, AEW has numerous advantages against LLE based approaches. For example, noise sensitivity of LLE can be alleviated by the parameterized form of the edge weights, and the similarity form for the edges weights is very reasonable for graph-based methods. We also provide several interesting properties of AEW, by which our objective function can be motivated analytically. We examined the performance of AEW by using two synthetic and eight real benchmark datasets. Experimental results demonstrated that AEW can improve the performance of the harmonic Gaussian field (HGF) model substantially, and we also saw that AEW outperformed LLE based approaches in all cases of real datasets except only one case.

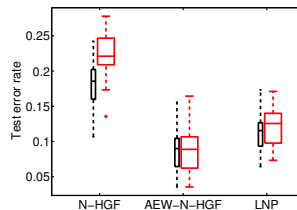

Figure 4: Comparison in test error rates of $k = 10$ and 20 (COIL $\ell = 50$). Two boxplots of each method correspond to $k = 10$ in the left (with a smaller width) and $k = 20$ in the right (with a larger width).

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
