[Supplementary Material]

# Supplementary Material for "Manifold-based Similarity Adaptation for Label Propagation"

**Masayuki Karasuyama and Hiroshi Mamitsuka**
Bioionformatics Center, Institute for Chemical Research
Kyoto University
Gokasyo, Uji, Kyoto, Japan
{karasuyama,mami}@kuicr.kyoto-u.ac.jp

## 1 Proof of Theorem 1

**Theorem 1.** *Suppose $\boldsymbol{x}_i$ can be approximated by its neighbors as follows*

$$\boldsymbol{x}_i = \frac{1}{D_{ii}} \sum_{j \sim i} W_{ij} \boldsymbol{x}_j + \boldsymbol{e}_i, \tag{1}$$

*where $\boldsymbol{e}_i \in \mathbb{R}^p$ represents an approximation error. Then, the same adjacency matrix reconstructs the output $y_i \in \mathbb{R}$ with the following error:*

$$y_i - \frac{1}{D_{ii}} \sum_{j \sim i} W_{ij} y_j = \boldsymbol{J} \boldsymbol{e}_i + O(\|\delta \boldsymbol{\tau}_i\|_2^2) + O(\varepsilon_x + \varepsilon_y), \tag{2}$$

*where $\boldsymbol{J} = \frac{\partial h(\boldsymbol{\tau}_i)}{\partial \boldsymbol{\tau}^\top} \left( \frac{\partial g(\boldsymbol{\tau}_i)}{\partial \boldsymbol{\tau}^\top} \right)^+$ with superscript $^+$ indicates pseudoinverse, and $\delta \boldsymbol{\tau}_i = \max_j (\|\boldsymbol{\tau}_i - \boldsymbol{\tau}_j\|_2^2)$.*

*Proof.* Let $\beta_j = W_{ij}/D_{ii}$ (Note that then $\sum_{j \sim i} \beta_j = 1$). Assuming that $g$ is smooth enough, we obtain the following first-order Taylor expansion at $\boldsymbol{\tau}_i$ for the right hand side of (1).

$$\boldsymbol{x}_i = \sum_{j \sim i} \beta_j \left( g(\boldsymbol{\tau}_i) + \frac{\partial g(\boldsymbol{\tau}_i)}{\partial \boldsymbol{\tau}^\top} (\boldsymbol{\tau}_j - \boldsymbol{\tau}_i) + O(\|\boldsymbol{\tau}_j - \boldsymbol{\tau}_i\|_2^2) \right)$$
$$+ \boldsymbol{e}_i + O(\varepsilon_x),$$

Arranging this equation, we obtain

$$\frac{\partial g(\boldsymbol{\tau}_i)}{\partial \boldsymbol{\tau}^\top} \sum_{j \sim i} \beta_j (\boldsymbol{\tau}_j - \boldsymbol{\tau}_i) = -\boldsymbol{e}_i + O(\|\delta \boldsymbol{\tau}_i\|_2^2) + O(\varepsilon_x).$$

If the Jacobian matrix $\frac{\partial g(\boldsymbol{\tau}_i)}{\partial \boldsymbol{\tau}^\top}$ has full column rank, we obtain

$$\sum_{j \sim i} \beta_j (\boldsymbol{\tau}_j - \boldsymbol{\tau}_i) = - \left( \frac{\partial g(\boldsymbol{\tau}_i)}{\partial \boldsymbol{\tau}^\top} \right)^+ \boldsymbol{e}_i + O(\|\delta \boldsymbol{\tau}_i\|_2^2)$$
$$+ O(\varepsilon_x). \tag{3}$$

On the other hand, we can see

$$\sum_{j\sim i}\beta_j y_j = \sum_{j\sim i}\beta_j \Big(h(\boldsymbol{\tau}_i) + \frac{\partial h(\boldsymbol{\tau}_i)}{\partial \boldsymbol{\tau}^\top}(\boldsymbol{\tau}_j - \boldsymbol{\tau}_i)$$

$$+ O(\|\boldsymbol{\tau}_j - \boldsymbol{\tau}_i\|_2^2)\Big) + O(\varepsilon_y)$$

$$= y_i + \frac{\partial h(\boldsymbol{\tau}_i)}{\partial \boldsymbol{\tau}^\top}\sum_{j\sim i}\beta_j(\boldsymbol{\tau}_j - \boldsymbol{\tau}_i)$$

$$+ O(\|\delta\boldsymbol{\tau}_i\|_2^2) + O(\varepsilon_y) \tag{4}$$

Substituting (3) into (4), we obtain

$$y_i - \sum_{j\sim i}\beta_j y_j = \frac{\partial h(\boldsymbol{\tau}_i)}{\partial \boldsymbol{\tau}^\top}\left(\frac{\partial g(\boldsymbol{\tau}_i)}{\partial \boldsymbol{\tau}^\top}\right)^+ \boldsymbol{e}_i$$

$$+ O(\|\delta\boldsymbol{\tau}_i\|_2^2) + O(\varepsilon_x + \varepsilon_y).$$

$\square$

## 2 Theorem 1 for Normalized Laplacian

**Theorem 2.** *Suppose that $\boldsymbol{x}_i$ can be approximated by its neighbors as follows*

$$\boldsymbol{x}_i = \sum_{j\sim i}\frac{W_{ij}}{\sqrt{D_{ii}D_{jj}}}\boldsymbol{x}_j + \boldsymbol{e}_i, \tag{5}$$

*where $\boldsymbol{e}_i \in \mathbb{R}^p$ represents an approximation error. Then, the same adjacency matrix reconstructs the output $y_i \in \mathbb{R}$ with the following error:*

$$y_i - \sum_{j\sim i}\frac{W_{ij}}{\sqrt{D_{ii}D_{jj}}}y_j = (1 - \sum_{j\sim i}\gamma_j)(h(\boldsymbol{\tau}_i) + \boldsymbol{J}g(\boldsymbol{\tau}_i))$$

$$+ \boldsymbol{J}\boldsymbol{e}_i + O(\|\delta\boldsymbol{\tau}_i\|_2^2) + O(\varepsilon_x + \varepsilon_y), \tag{6}$$

*where*

$$\gamma_j = \frac{W_{ij}}{\sqrt{D_{ii}D_{jj}}}.$$

*Proof.* The proof is almost the same as Theorem 1. However, the sum of the coefficients $\gamma_j$ (corresponding to $\beta_j$ in Theorem 1) cannot be 1. Applying the same Taylor expansion to the right hand side of (5), we obtain

$$\frac{\partial g(\boldsymbol{\tau}_i)}{\partial \boldsymbol{\tau}^\top}\sum_{j\sim i}\gamma_j(\boldsymbol{\tau}_j - \boldsymbol{\tau}_i) = -\boldsymbol{e}_i + (1 - \sum_{j\sim i}\gamma_j)g(\boldsymbol{\tau}_i)$$

$$+ O(\|\delta\boldsymbol{\tau}_i\|_2^2) + O(\varepsilon_x).$$

On the other hand, applying Taylor expansion to $y_i - \sum_{j\sim i}\gamma_j y_j$, we obtain

$$y_i - \sum_{j\sim i}\gamma_j y_j = (1 - \sum_{j\sim i}\gamma_j)h(\boldsymbol{\tau}_i) - \frac{\partial h(\boldsymbol{\tau}_i)}{\partial \boldsymbol{\tau}^\top}\sum_{j\sim i}\gamma_j(\boldsymbol{\tau}_j - \boldsymbol{\tau}_i)$$

$$+ O(\|\delta\boldsymbol{\tau}_i\|_2^2) + O(\varepsilon_y)$$

Using the above two equations, we obtain (6). $\square$