[Reviews · NeurIPS 2013]

Submitted by Assigned_Reviewer_4

This paper addresses the problem of assigning weights to edges in a graph for label propagation. The assumption is that the graph structure (and training labels) are provided. The proposed method is simple to implement and appears to perform well against standard approaches.

* Quality: The paper is technically sound and the claims are reasonably supported through experimental comparison. While many additional experiments could be run, I feel the evaluation is adequate.

* Clarity: The paper is generally well written. Many of the data points in Figure 3 are overlapping making it hard to distinguish between plots. Perhaps plotting on a different scale (log-scale) or relative to one of the baselines will help separate the curves. Minor point: x \in R^p on line 094 should be x_i \in R^p (i.e., include the _i index).

* Originality: Numerous works have considered the problem of estimating edge weights. The proposed approach shares similarities with some of these methods (e.g., LLE and diffusion maps). Nevertheless the objective is novel and appears to address some of the shortcomings of these methods.

* Significance: The proposed method is simple but performs well. It is likely to be useful for graph-based approaches.

Summary: This is a well written paper that addresses an important problem for graph-based learning.

Submitted by Assigned_Reviewer_5

The manuscript a new method “adaptive edge weighting” (AEW) that weights to the edges of a neighborhood graph for use with label propagation. The neighborhood graph is constructed using k-NN in the feature space, and then the edge weights are optimized based on their ability to reconstruct the feature vector associated with each node. The objective function is the same as LLE but the weights are derived from Gaussian kernels and the only parameters are the bandwidths of the kernels in each dimension, thereby forcing the weights to be positive and greatly reducing the number of free parameters. They call this procedure “feature propagation”. The manuscript contains a theorem that connects the reconstruction error in feature propagation (and the maximum neighbor distance) with the expected performance in the label propagation task. When tested on simulated data or eight classification / multiclass benchmarks and paired with one flavor of label propagation (Harmonic Gaussian Fields), AEW performs better than a version that does not optimize bandwidths or a previous, similar method (LNP) that uses the unrestricted LLE objective function to set the edge weights. The manuscript also points out that the method is less sensitive to increases in k when constructing the neighborhood graph because of the Gaussian kernel derived edge weights.

Clarity: The manuscript is well-organized and easy to follow. The experimental section is clear although one very minor issue is that I am not sure that I could perfectly reproduce their results, as the exact optimization procedure they use for the bandwidths is not described (steepest descent? CG?) and the synthetic data set is not provided.

Quality: This is a well-constructed, good quality paper – it provides a new methodology, some insight into why this method is appropriate through the aforementioned theorem, as well as sufficient experimental evaluation. I have some questions about the validation. LNP appears to use a different neighborhood graph than AEW, how much of the performance improvement of AEW is due to the different graphs? Also, AEW doesn’t generate a normalized Laplacian (does LNP?), but normalized version of the AEW Laplacian was used in the validations, presumably to improve performance. This issue is only discussed briefly; I think some more discussion is warranted.

Originality: The contribution seems original although is a slight, but crafty, modification of a previous technique (LNP) for setting the edge weights.

Significance: The LNP paper has >200 Google scholar citations and the described method performs better, is relatively easy to implement, and has no tuning parameters. Given a relevant learning problem, this would be the first method that I would try. So, it does have some significance. On the other hand, this does seem like an incremental improvement on well-studied problem although the theorem does provide a basis on which to develop other “feature propagation” based weighting schemes. One other issue that could be addressed in the introduction to increase the significance of the work, is why one would want to convert feature vectors into a graph rather than working directly in the feature space.

Minor issues:
Line 119: Are the limits of the argmax correct here? “argmaxj≤j Fij “
Summary: A new method for setting edge weights in a neighborhood graph to be used for label propagation. It works well, has some theoretical support, and no tuning parameters once provided with the neighborhood graph.

Submitted by Assigned_Reviewer_6

A method for improving the similarity metric used for edge weights in
semi-supervised learning is presented. The authors first discuss
standard methods of semi-supervised learning--graph construction and
label-propagation. Discussion and references to prior work
demonstrate that the assignment of edge weights is a critical yet
unsolved problems.

The authors then propose a learned weighting of the metric in a
standard Gaussian-style kernel for edge weights. The weighting
optimization criterion is based on reconstruction of a feature vector
from it's neighbors (as in the harmonic case for function values).
The authors motivate feature propagation as method of ensuring good
label (output value) propagation.

The authors then evaluate their method on a variety of synthetic and
real data sets. Results show the new method is promising.

Quality
The results of the paper demonstrate the value of the technique
presented. Optimizing the weighting in the Gaussian in (1) is a
simple approach to improving performance, and the experiments validate
it is important. A slight weakness is that although intuitively the
feature propagation and harmonic-style prediction from neighbors is
compelling, the explanation in Section 4 is hard to follow. Is there
a way to make this more compelling and crisp?

Clarity
The methods in the paper are clearly presented. Notation and
experimental descriptions are well done.

Originality
This paper has good references to and builds upon prior
work. Some prior work has looked at optimization of kernels by metric
reweighting, but this is not in the context of semi-supervised
learning. The novelty in this case appears to be the combination of
weighting with the semi-supervised learning task.

Significance
The method is compelling and the area of investigation,
semi-supervised learning and graphs, continues to be an important
topic.


Summary: The authors present a simple yet effective method for modifying edge weights in semi-supervised learning. Algorithms and methods are presented which produce compelling semi-supervised learning results on synthetic and real data sets.
Author Feedback

Author feedback is not available.